# Dietary Factors and Risks of Cardiovascular Diseases: An Umbrella Review

**DOI:** 10.3390/nu12041088

**Published:** 2020-04-15

**Authors:** Kridsada Chareonrungrueangchai, Keerati Wongkawinwoot, Thunyarat Anothaisintawee, Sirimon Reutrakul

**Affiliations:** 1Department of Family Medicine, Faculty of Medicine, Ramathibodi Hospital, Mahidol University, Praram VI Road, Rachathevee, Bangkok 10400, Thailand; thonsmn@gmail.com (K.C.); miupang3456@gmail.com (K.W.); 2Department of Clinical Epidemiology and Biostatistics, Ramathibodi Hospital, Mahidol University, Praram VI Road, Rachathevee, Bangkok 10400, Thailand; 3Division of Endocrinology, Diabetes and Metabolism, University of Illinois College of Medicine at Chicago, 835 S Wolcott, Ste E625, Chicago, IL 60612, USA; sreutrak@uic.edu

**Keywords:** dietary factor, cardiovascular disease, umbrella review

## Abstract

Unhealthy diet is a significant risk factor for cardiovascular diseases (CVD). Therefore, this umbrella review aims to comprehensively review the effects of dietary factors, including dietary patterns, food groups, and nutrients on CVD risks. Medline and Scopus databases were searched through March 2020. Systematic reviews with meta-analyses (SRMA) of randomized controlled trials (RCTs) or observational studies measuring the effects of dietary factors on CVD risks were eligible. Fifty-four SRMAs, including 35 SRMAs of observational studies, 10 SRMAs of RCTs, and 9 SRMAs of combined RCT and observational studies, were included for review. Findings from the SRMAs of RCTs suggest the significant benefit of Mediterranean and high-quality diets for lowering CVD risk, with pooled risk ratios (RRs) ranging from 0.55 (95%CI: 0.39–0.76) to 0.64 (95%CI: 0.53–0.79) and 0.70 (95%CI: 0.57–0.87), respectively. For food nutrients, two SRMAs of RCTs found that high intake of n-3 polyunsaturated fatty acid (PUFA) significantly reduced CVD risks, with pooled RRs ranging from 0.89 (95%CI: 0.82, 0.98) to 0.90 (95%CI: 0.85–0.96), while evidence of efficacy of n-6 PUFA and combined n-3 and n-6 PUFA were inconsistent. Moreover, results from the SRMAs of RCTs did not find a significant benefit of a low-salt diet and low total fat intake for CVD prevention. For food groups, results from the SRMAs of cohort studies suggest that high intakes of legumes, nuts, and chocolate, as well as a vegetarian diet significantly reduced the risk of coronary heart disease, with pooled RRs of 0.90 (95%CI: 0.84–0.97), 0.68 (95%CI: 0.59–0.78), 0.90 (95%CI: 0.82–0.97), and 0.71 (95%CI: 0.57–0.87), respectively. Healthy dietary patterns had a significant benefit for CVD prevention. With the substitutional and synergistic interactions between different food groups and nutrients, dietary recommendations for CVD prevention should be focused more on healthy dietary patterns than single food groups or nutrients.

## 1. Introduction

Cardiovascular diseases (CVD), today’s leading causes of death, accounts for one third of all mortality worldwide [1]. In the past decade, CVD mortalities have increased globally by 12.5% [2]. One significant risk factor of CVD is an unhealthy diet, which is also related to other CVD risk factors, such as hypertension, diabetes mellitus (DM), and obesity [3,4]. Therefore, encouraging healthy diet adherence is important in decreasing CVD morbidity and mortality.

CVD dietary factors is usually classified into three main types: dietary patterns (e.g., the Mediterranean diet and the Dietary Approaches to Stop Hypertension (DASH) diet), food groups (e.g., fruits, vegetables, nuts, whole grains, and legumes), and food nutrients (e.g., sodium, saturated fat, and monounsaturated fat). However, most evidence has focused on dietary fats, due to the established relationship between serum cholesterol level and CVD risks. Previous evidence on the association between dietary fat intake and CVD prevention is inconsistent and is still being debated. For instance, in 2017 the American Heart Association (AHA) recommended lowering saturated fat intake and replacing it with unsaturated fat, especially polyunsaturated fatty acids (PUFA), for CVD prevention [5]. However, some systematic reviews and meta-analyses (SRMA) of randomized controlled trials (RCT) did not show a significant benefit of PUFA for reducing CVD risks [6,7], and the findings from an 18-country cohort study also concluded that “total fat and types of fat were not associated with CVD” [8]. Similarly, findings from SRMAs [9,10,11,12] of the effects of other dietary factors, such as vegetables, fruits, and fibers on CVD risks were conflicting, demonstrating the complexity of the link between diets and CVD pathogenesis.

Humans usually have dietary patterns that are a combination of multiple diets composed of multiple nutrients that have synergistic interactions. Hence, to understand the association between diets and CVD risk, we must consider all nutrients, food groups, and dietary patterns, as well as the interrelationship between them. Many SRMAs measuring the effects of dietary factors and CVD risks have been published over the past decade [6,7,9,10,11,12]. However, findings from these SRMAs are mostly conflicting. Therefore, to comprehensively summarize the effects of dietary factors on CVD risks, the strength, precision, and potential bias of the findings from previous SRMAs should be explored.

An umbrella review is tertiary research that provides a comprehensive overview of evidence from SRMAs [13]. Hence, this type of review can reveal the strength and precision of the effect estimates and explore the potential bias of previous SRMAs. Therefore, this umbrella review aims to comprehensively review the evidence regarding the effects of nutrients, food groups, and dietary patterns on CVD risks. The effects of each dietary factors on subtypes of CVD, including coronary heart disease (CHD), stroke, CVD mortality, and all-cause mortality were also explored. Moreover, the potential bias and the consistency of evidence from the previous SRMAs of RCTs and observational studies were investigated.

## 2. Materials and Methods

This umbrella review was conducted according to the preferred reporting items for systematic reviews and meta-analyses (PRISMA) guidelines [14]. The review protocol was registered in PROSPERO (CRD42018105292).

### 2.1. Literature Search and Study Selection

Medline and Scopus databases were searched from their inceptions to March 2020 to identify the relevant studies. Search terms and strategies of each database are presented in the Appendix A. Two reviewers (K.C. and K.W.) independently selected the studies. Disagreement between two reviewers were decided by consensus with the third party (T.A.). Systematic reviews with meta-analyses of observational studies or RCTs were eligible, if they met the following criteria; (1) the study’s participants were from the general population or were people with high risks for CVD; (2) interested interventions or exposures were dietary factors; (3) the outcomes of interest were CVD, or all-cause mortality; and (4) the pooled risk ratios (RR) or odds ratios (OR), in accordance with their 95% confidence intervals (CI) for dietary factors/diet interventions and outcomes, were reported. Studies were excluded if they included only CVD patients as participants.

### 2.2. Data Extraction

The following information was extracted from each SRMA: (1) characteristics of eligible SRMAs, including first authors, year of publication, country of corresponding authors, sources of funding support, conflict of interest (COI), types of participants, interested exposures, interventions, comparisons, outcomes, and numbers of primary studies included in SRMA; (2) results of meta-analysis, including pooled RRs or ORs, and their 95%CIs for high versus low, as well as dose response meta-analyses, degree of heterogeneity, and publication bias. The data of primary studies included in each SRMA (i.e., mean age and total numbers of study’s participants, percentage male, and study settings) were also extracted. Two reviewers (K.C. and K.W.) extracted the data, and the data were validated by the third reviewer (T.A.).

### 2.3. Methodological Quality Assessment

The methodological quality of included SRMAs were assessed using the Assessing the Methodological Quality of Systematic Review (AMSTAR) 2. AMSTAR 2 has 16 items in total, including reporting review questions according to Population, Intervention, Comparator, Outcome, protocol registration, study selection, literature search, data extraction, risk of bias assessment, sources of funding, methods of meta-analysis, using risk of bias assessment for data analysis and interpretation, reporting of heterogeneity, publication bias, and conflict of interest. The items were classified as critical and non-critical domains. Overall confidence in the results of the SRMA was rated as high, moderate, low, or critically low confidence, if the SRMA answered “yes” in 0–1 items of a non-critical domain, >1 items in a non-critical domain, 1 item in a critical flaw domain with/without a non-critical domain, or >1 items in a critical flaw domain with/without non-critical domain, respectively.

### 2.4. Dietary Factors or Interventions

Dietary factors were classified as (1) dietary patterns, (2) food groups, and (3) food nutrients. Dietary patterns referred to the combination of different foods, beverages, and nutrients, and the frequency with which they are routinely consumed [15], such as the Mediterranean diet, the DASH diet, a high quality diet as measured by Healthy Eating Index (HEI) or Alternate Healthy Eating Index (AHEI) scores, or a diet with a low glycemic index. Each food group is defined as a compilation of foods with similar nutritional properties; the food groups were divided into (1) dairy products; (2) fruits; (3) vegetables; (4) meat; (5) grains, beans, and legumes; (6) oils; (7) confections (e.g., sugar-sweetened beverages and chocolate); and (8) coffee. Nutrients, such as protein, fat, carbohydrates, fiber, vitamins, and minerals, are chemical compounds that are used by human bodies to preserve health [16].

Diet interventions referred to any modification or treatment on an individual’s diet with a prepared goal [17]. These interventions could be provided by diet supplements or education only.

### 2.5. Outcomes of Interest

The outcomes of interest were all-cause mortality and cardiovascular diseases. Cardiovascular diseases were defined as cardiovascular mortality; coronary heart diseases (CHD), including acute myocardial infarction (MI); stable and unstable angina; and cerebrovascular disease (CVA), including hemorrhagic and ischemic strokes.

### 2.6. Data Analysis

Characteristics of included SRMAs were described qualitatively. The pooled effect size of each dietary factor and interventions for each CVD outcome were summarized qualitatively. Heterogeneity between studies and publication bias for each pooling were also presented. Pooled effect sizes of each dietary factor are presented in forest plots, since we could not include the results from all included studies. If there were more than one systematic review and meta-analyses that investigated the effect size of the similar type of dietary factor, pooled risk ratios from SRMAs of RCTs with the highest quality from AMSTAR 2 were selected to present in the forest plots. If there were no SRMAs of RCTs, pooled risk ratios from SRMAs of observational studies with the highest quality according to AMSTAR 2 were selected to present in the forest plots instead.

## 3. Results

The results of study selection and reasons for exclusion are presented in Figure 1. Fifty-four SRMAs met the inclusion criteria and were eligible for review. Characteristics of included SRMAs are presented in Appendix A. Almost all SRMAs (48/54) were published after the year 2010. Eighteen SRMAs (33.33%) were conducted in European countries, followed by Asian countries (31.48%), the United Kingdom (16.67%), and the United States (7.41%). Two, three, and one SRMAs were conducted in East Asia, Australia and New Zealand, and South America, respectively. Six SRMAs had a COI with food industries, forty-one SRMAs reported no COI, and four SRMAs did not state anything about a COI. Thirty-five SRMAs (65%) included only observational studies, 10 SRMAs (19%) included only RCTs, and nine SRMAs (17%) included both observational studies and RCTs. In addition, 15, 10, 14, and 5 SRMAs featured dietary patterns, food groups, food nutrients, and both food groups and nutrients, respectively. Lastly, seven, two, and one SRMAs featured diet interventions in food nutrients, dietary patterns, and both food nutrients and dietary patterns, respectively [17].

### 3.1. Dietary Patterns

A total of 18 SMRAs, (13 SRMAs of observational studies, two SRMAs of RCTs and observational studies, and three SRMAs of RCTs) assessed the effects of dietary patterns and CVD risks. For SRMAs of observational studies, dietary patterns were Mediterranean diet (eight SRMAs), DASH diet (three SRMAs), diets with high HEI and AHEI scores (two SRMAs), HEI/AHEI and cardiovascular health (CVH) scores (one SRMA), and Diet Inflammatory Index (DII) scores (one SRMA). For SRMAs of RCTs, the interventions were prescribing a Mediterranean diet (one SRMA) [18] and modifying diet quality by lowering the consumption of carbohydrates, fat, and calories, and increasing the consumption of fish, vegetables, complex carbohydrates, and fiber (two SRMA) [17,19]. Most of the SRMAs (16/18) considered the general population as the study’s participants, while two included only high-risk populations (e.g., patients with obesity, hypertension, and DM). The mean age and percentage of male participants ranged from 18 to 104 years, and 0% to 100%, respectively (see Appendix A). The effects of each dietary pattern are described in Figure 2A–D and Appendix A.

#### 3.1.1. Mediterranean Diet

For all-cause mortality, two SRMAs of observational studies [20,21] found that adherence to Mediterranean diet significantly decreased risk of all-cause mortality in the general population, with a pooled RR ranging from 0.91 (95%CI: 0.89–0.94) to 0.92 (95%CI: 0.90–0.94). However, a finding from an SRMA of RCTs [18] suggested a non-significant benefit of the Mediterranean diet in reducing all-cause mortality in high-risk populations (pooled RR = 1.00; 95%CI: 0.86–1.15; see Figure 2A).

For CVD mortality, both SRMAs of observational studies and RCTs [20,22] found a significant benefit of the Mediterranean diet in decreasing CVD mortality in the general population, with pooled RRs ranging from 0.59 to 0.91 (see Figure 2B), while the SRMA of RCTs [18] found no significant effect in high-risk populations (pooled RR = 0.90; 95%CI: 0.72–1.11; see Appendix A).

Four [18,22,23,24], three [18,22,24], and six [18,22,24,25,26,27] SRMAs reported the outcomes as CVD, CHD, and stroke, respectively. Both the SRMAs of observational studies and RCTs found the significant benefit of a Mediterranean diet in reducing the risk of CVD (pooled RRs ranging from 0.55 to 0.81), CHD (pooled RRs ranging from 0.65 to 0.72), and stroke (pooled RRs ranging from 0.64 to 0.84) in both general and high-risk populations (see Figure 2C–D).

#### 3.1.2. DASH Diet

Three SRMAs of observational studies [28,29,30] assessed the effect of a DASH diet on CVD risk. These SRMAs found that high adherence to a DASH diet significantly decreased the risk of CHD (pooled RRs ranging from 0.79 (95%CI: 0.71–0.88) to 0.95 (95%CI: 0.94–0.97)) and stroke (pooled RRs ranging from 0.81 (95%CI: 0.72–0.92) and 0.88 (95%CI: 0.83–0.93; see Figure 2D and Appendix A).

#### 3.1.3. Diet Quality

Diet quality was measured by HEI/AHEI (two SRMAs) [31,32], HEI/AHEI and CVH (one SRMA) [33], and DII (one SRMA) [34] scores. The effects of diet quality on CVD risk are presented in Appendix A. High HEI/AHEI and CVH scores reflect the high quality of diet, whereas high DII scores reveal a poor-quality diet. Participants of all SRMAs were the general population. All three SRMAs of observational studies have consistent findings that consuming diets with high HEI/AHEI and CVH scores significantly decreased the risk of all-cause mortality (pooled RRs ranging from 0.54 to 0.78), CVD mortality (pooled RRs ranging from 0.30 to 0.77), and CVD (pooled RR = 0.78). However, diets with a high DII score showed a significantly increased risk of CVD (pooled RR = 1.35; 95%CI: 1.11–1.63) and CVD mortality (pooled RR = 1.37; 95%CI: 1.11–1.70; see Appendix A).

Evidence from the SRMA of RCTs suggests that increasing high-quality diet consumption in the high-risk population significantly decreased CVD risk [19] (pooled RR = 0.70; 95%CI: 0.57–0.87; see Figure 2C). However, there was no significant effect of increasing high-quality diet consumption in lowering all-cause (pooled RR = 0.97; 95%CI: 0.92–1.04) and CVD mortality (pooled RR = 0.91; 95%CI: 0.82–1.02) in the general population [17] (see Figure 2A–B).

### 3.2. Food Groups

A total of 14 SRMAs (10 SRMAs of observational studies and four SRMAs of both observational studies and RCTs) assessed the effects of food groups on CVD risks. The food groups considered were (1) fruits and vegetables; (2) nuts, whole grains, and legumes; (3) fish; (4) a vegetarian diet; (5) olive oil; (6) chocolate; (7) coffee; and (8) green tea. Participants of all 14 SRMAs were from the general population. Mean age, male percentage, and total number of participants ranged from 20 to 100 years, 0% to 100%, and 51 to 454,775, respectively. The effects of each food group are presented in Figure 3 and Appendix A.

#### 3.2.1. Fruits and Vegetables

Three SRMAs of observational studies assessed the association between fruit and vegetable intake and CVD risks [10,11,12]. High fruit and vegetables intake significantly decreased the risk of stroke (pooled RR ranging from 0.77 (95%CI: 0.71–0.84) to 0.86 (95%CI: 0.79–0.93)) [11] (Figure 3A), but did not decrease the risk of CHD (pooled RR ranging from 0.82 (95%CI: 0.66–1.02) to 0.86 (95%CI: 0.71–1.05)) [10] (Figure 3B), and CVD mortality (pooled RRs ranging from 0.95 (95%CI: 0.89–1.02) to 0.96 (95%CI: 0.83–1.11) [12] (Figure 3C).

#### 3.2.2. Vegetarian Diet

One SRMA of observational studies investigated the association between vegetarian diet and CVD risks [35]. High adherence to vegetarian diet significantly lowered the CHD risk, with a pooled RR of 0.71 and 95%CI: 0.57–0.87 (Figure 3B), but did not lower CVD mortality (pooled RR = 0.87; 95%CI: 0.68–1.11) and risk of stroke (pooled RR = 0.93; 95%CI: 0.70–1.23; see Figure 3A,C).

#### 3.2.3. Nuts, Whole Grains, and Legumes

Two SRMAs each of observational studies assessed the effect of nuts [36,37], legumes [12,38], and whole grains [39,40]. Both high vs. low and dose response analyses of nut intake demonstrated a significantly beneficial effect on the risk of CHD, with pooled RRs ranging from 0.68 (95%CI: 0.59–0.78) to 0.90 (95%CI: 0.87–0.94) [37] (Figure 3B), and stroke, with pooled RRs ranging from 0.86 (95%CI: 0.79–0.94) to 0.88 (95%CI: 0.80–0.97) [36] (Figure 3A). Both high vs. low and dose response analyses of whole grain intake indicated a significant effect in lowering all-cause (pooled RRs ranging from 0.81 (95%CI: 0.76–0.85) to 0.87 (95%CI: 0.84–0.90)) and CVD mortality (pooled RRs ranging from 0.66 (95%CI: 0.56–0.67) to 0.81 (95%CI: 0.74–0.89)) [39,40] (see Figure 3C,D). In addition, a high intake of whole grain significantly decreased risk of CHD, with a pooled RR of 0.80 (95%CI: 0.70–0.91), but did not reduce risk of stroke (pooled RR = 0.86; 95%CI: 0.61–1.21). High legume intake also significantly reduced CVD mortality (pooled RR = 0.89; 95%CI: 0.82–0.98) [12] and CHD risk (pooled RR = 0.90; 95%CI: 0.84–0.97) [38] (see Figure 3B,C), but not risk of stroke (pooled RR = 1.01; 95%CI: 0.89–1.14) [38] (see Figure 3A).

#### 3.2.4. Fish

Two SRMAs of observational studies assessed the association between fish intake and CVD risks [6,41], and found that high fish intake significantly reduced CVD mortality, with pooled RRs ranging from 0.75 (95%CI: 0.62–0.92) to 0.82 (95%CI: 0.71, 0.94; see Figure 3C). However, the results of CHD risk were inconsistent between the two SRMAs, as Whelton et al. [41] show a significant benefit of high fish intake (pooled RR = 0.83; 95%CI: 0.69–0.99), while Skeaff et al.’s results [6] are non-significant (pooled RR = 0.87; 95%CI: 0.71–1.06).

#### 3.2.5. Olive Oil

Findings from one SRMA of observational studies and RCTs indicate that high olive oil consumption [42] significantly decreases the risk of stroke (pooled RR = 0.76; 95%CI: 0.67–0.86; see Figure 3A). There was no significant effect of olive oil on CHD risk (pooled RR = 0.94; 95%CI: 0.78–1.14; see Figure 3B).

#### 3.2.6. Chocolate

Two SRMAs of observational studies investigated the association between chocolate consumption and CVD risks [43,44]. The results demonstrate that high chocolate consumption significantly decreased the risk of CHD (pooled RRs ranging from 0.71 (95%CI: 0.56–0.92) to 0.90 (95%CI: 0.82–0.97)), stroke (pooled RRs ranging from 0.79 (95%CI: 0.70–0.87) to 0.84 (95%CI: 0.78–0.90)), and CVD mortality (pooled RR = 0.55 (95%CI: 0.36–0.83; see Figure 3A–C).

#### 3.2.7. Coffee and Green Tea

One SRMA of observational studies found that drinking one cup of coffee per day significantly reduced all-cause and CVD mortality, with pooled RRs of 0.92 (95%CI: 0.91–0.94) and 0.89 (95%CI: 0.86–0.91), respectively [45] (see Figure 3C,D). For green tea, consuming 1–3 cups of green tea per day significantly lowers the risk of stroke (pooled RR = 0.64 (95%CI: 0.47–0.86), while the risk of CVD, all-cause, and CVD mortality were not significantly different between green tea intake and non-intake [46].

### 3.3. Food Nutrients

Sixteen SRMAs of observational studies, seven SRMAs of RCTs, and one SRMA of combined observational studies and RCTs investigated the effect of food nutrients on CVD risk in the general population. Mean age, percentage male, and total number of study’s participants ranged from 20–89 years, 0–100%, and 16–388,229, respectively. Most SRMAs studied fat intake (11/24), followed by fiber (5/20), sodium (3/20), flavonoid (3/20), potassium (1/20), and calcium intakes (1/20). Effect of each food nutrients are presented in Figure 4 and Figure 5, and Appendix A.

#### 3.3.1. Fat Intake

Fat intake was classified as (1) total fat intake, (2) saturated fatty acid (SFA), (3) MUFA, (4) n-3 PUFA, (5) n-6 PUFA, and (6) trans fatty acid (TFA). For total fat intake, the results from SRMAs of observational studies found that a high total fat intake did not significantly increase risk of CVD mortality (pooled RRs ranging from 0.94 (95%CI: 0.74–1.18) to 1.04 (95%CI: 0.98–1.10)) [6,47] (Figure 4C) and CHD (pooled RR = 0.93; 95%CI: 0.84–1.03) [6] (Figure 5B). Evidence from SRMAs of RCTs also indicated that modification of the amount of total fat intake did not significantly decrease risk of all-cause mortality (pooled RRs ranging from 0.98 (95%CI: 0.86–1.12) to 0.99 (95%CI: 0.94–1.05)) [48,49] (Figure 4B), CVD mortality (pooled RRs ranging from 0.91 (95%CI: 0.77–1.07) to 1.00 (95%CI: 0.80–1.24)) [6,48,49], or CHD (pooled RR = 0.93 (95%CI: 0.84–1.04) [6] (Figure 5C) in the general population.

Findings from two SRMAs of observational studies showed that high SFA intake was not significantly associated with risk of CVD mortality [6,47] and CHD [6] (see Figure 4B and Figure 5B). However, one SRMA of observational studies found that a high SFA intake was significantly associated with lower risk of ischemic stroke (pooled RR = 0.89; 95%CI: 0.82–0.96) [50] (see Figure 5A).

One meta-analysis of cohort studies assessed the association between high trans-fat intake and CVD risk, and found that high trans-fat intake significantly increased the risk of CVD mortality (pooled RR = 1.32; 95%CI: 1.08–1.61; Figure 4C) and CHD (pooled RR = 1.25; 95%CI: 1.07–1.46; Figure 5B).

Findings from one SRMA of observational studies suggest that high MUFA intake significantly decreased the risk of all-cause mortality (pooled RR = 0.89; 95%CI: 0.83–0.96) (Figure 4A) and stroke (pooled RR = 0.83; 95%CI: 0.71–0.97) (Figure 5A). However, the effects of MUFA on CVD mortality were inconsistent between two SRMAs. Schwingshackl et al. show that high MUFA intake significantly reduced the risk of CVD mortality (pooled RR = 0.88; 95%CI: 0.80–0.96) [51], while Skeaff et al. show a non-significant effect (pooled RR = 0.85; 95%CI: 0.60–1.20) [6]. However, both SRMAs found that high MUFA intake did not significantly reduce CHD risk (see Figure 5B).

PUFA is classified as n-3 PUFA, n-6 PUFA, and combined n-3 and n-6 PUFA. Three SRMAs of RCTs assessed the effect of n-3 PUFA on CVD risk. Two SRMAs found that n-3 PUFA significantly lowered the risk of CVD (pooled RR ranging from 0.89 (95%CI: 0.82-0.98) to 0.90 (95%CI: 0.85-0.96)) [6,52], while one SRMA found a non-significant effect of n-3 PUFA (pooled RR = 0.99; 95%CI: 0.94-1.04) [53]. However, all-cause mortality rate was not significantly different between n-3 PUFA and placebo groups [52] (see Figure 4B). The findings about the effect of n-3 PUFA on CVD mortality were inconsistent between these two SRMAs, as the results of Delgado-Lista et al. suggest a significant benefit (pooled RR = 0.91; 95%CI: 0.83–0.99) [52], while results of Skeaff et al. [6] and Abdelhamid et al. [53] indicated a non-significant effect (pooled RRs ranging from = 0.88; 95%CI: 0.76–1.01 to 0.99; 95%CI: 0.94–1.04). In addition, the results from Abdelhamid et al. indicated that n-3 PUFA significantly decreased the risk of CHD, but the benefit of n-3 PUFA was not seen for stroke prevention. 

One meta-analysis of RCTs indicated a non-significant benefit of n-6 PUFA for the prevention of all-cause mortality (Figure 4A), while there were inconsistent findings for the outcomes on CVD mortality and CHD risk. One meta-analysis of RCTs found that high n-6 PUFA intake significantly decreased the risk of CVD mortality (pooled RR = 0.81; 95%CI: 0.70–0.95) [54], while another meta-analysis of RCT found a non-significant effect (pooled RR = 1.17; 95%CI: 0.82–1.68) [7]. Two meta-analyses of cohort studies assessing the effect of n-6 PUFA on the risk of CHD also show conflicting results, with Farvid et al. suggesting the significant benefit of high n-6 intake for prevention of CHD (pooled RR = 0.81; 95%CI: 0.70–0.95) [54], and Skeaff et al. demonstrating a non-significant benefit (pooled RR = 1.05; 95%CI: 0.92–1.20) [6]. For combined n-3 and n-6 PUFA, the results from two SRMAs of RCTs [6,7] indicate that high PUFA intake was not significantly associated with CHD risk, all-cause, and CVD mortality.

#### 3.3.2. Fiber

Five SRMAs of observational studies assessed the association between fiber intake and CVD risk. Both high vs. low and dose response analyses from all SRMAs suggest that high fiber intake significantly reduced the risk of all-cause mortality (pooled RR = 0.85; 95%CI: 0.79–0.91), CVD mortality (pooled RRs ranging from 0.69 (95%CI: 0.60–0.81) to 0.91 (95%CI: 0.88–0.94); Figure 4C) [12,40], stroke (pooled RRs ranging from 0.78 (95%CI: 0.69–0.88) to 0.88 (95%CI: 0.79–0.97); Figure 5A) [9,40,55], and CHD (pooled RRs ranging from 0.76 (95%CI: 0.69–0.83) to 0.86 (95%CI: 0.79–0.95); Figure 5B) [10,40].

#### 3.3.3. Sodium

One SRMA of RCTs, one SRMA of cohort studies, and one SRMA of observational studies and RCTs found that low sodium intake did not significantly reduce risk of CVD, all-cause mortality, and CVD mortality [56,57]; however, meta-analysis of cohort studies suggested that low sodium intake could significantly decrease the risk of stroke, with pooled RRs ranging from of 0.81 (95%CI: 0.70–0.93) to 0.94; 95%CI: 0.90–0.98 [57,58] (see Figure 5A).

#### 3.3.4. Flavonoid

Two SRMAs of observational studies found that the risk of CVD mortality and stroke in people having a high flavonoid intake was significantly lower than those with a low intake, with pooled RRs of 0.86 (95%CI: 0.75–0.98; Figure 4C) and 0.80 (95%CI: 0.65–0.98; Figure 5A) [59,60]. However, the risk of all-cause mortality was not significantly different between the two groups [59] (see Figure 4A).

One SRMA of observational studies assessed the effect of anthocyanins, which are a subtype of flavonoid, on CVD risk [61]. This SRMA found that a high intake of anthocyanins significantly decreased the risk of CVD mortality and CHD, with pooled RRs of 0.92 (95%CI: 0.87–0.97) and 0.91 (95%CI: 0.83–0.99), respectively (see Appendix A). However, the risk of stroke did not significantly decrease in people with a high intake of anthocyanins. 

#### 3.3.5. Potassium

One SRMA of observational studies assessed the association between potassium intake and CVD risk [62]. High potassium intake significantly decreased the risk of stroke (pooled RR = 0.79; 95%CI: 0.68–0.90; Figure 5A), while risk of CHD was not significantly different between high and low potassium intake (pooled RR = 0.92; 95%CI: 0.81–1.04; Figure 5B).

#### 3.3.6. Calcium

One meta-analysis of cohort studies assessed the effect of high calcium intake and CVD mortality [63], and found that a high calcium intake from diet and supplements did not significantly reduce the risk of all-cause mortality (pooled RR = 0.83; 95%CI: 0.70–1.00) or CVD mortality (pooled RR = 0.97; 95%CI: 0.89–1.07) (see Figure 4A,C).

### 3.4. Methodological Quality Assessment

Results of quality assessment are presented in Appendix A. Forty-eight out of 54 studies were classified as critically low confidence, five studies were classified as low confidence, and one study was classified as high confidence, according to AMSTAR-2 criteria. All of the studies having critically low confidence did not registered the review protocols, and most of those studies (90%) did not provide a list of excluded studies and reasons for exclusion. Around 50% of included studies did not consider the results of the risk of bias assessment in individual studies when interpreted, nor discussed the results. Most of the studies (51/54) used more than two databases for searching relevant studies, but only nine studies searched in databases for grey literatures. Most of the studies used appropriated methods of meta-analysis (52/54) and reported publication bias (45/54).

## 4. Discussion

This umbrella review provides a comprehensive summary of evidence about the effect of dietary factors on the risk of CVD. Evidence from RCTs and observational studies confirms the benefit of healthy dietary patterns, especially the Mediterranean diet, for the prevention of CVD, but the benefit for lowering all-cause and CVD mortality were inconsistent. The DASH diet also exhibited the ability to prevent CVD, but was only supported by observational studies.

A high intake of whole grains, legumes, fish, chocolate, and drinking one cup of coffee per day significantly decreased the risk of CVD mortality; a high intake of legumes, nuts, and chocolate, as well as a vegetarian diet could reduce risk of CHD. In addition, high intakes of vegetables and fruits, olive oil, and nuts, as well as dinking 1–3 cups of green tea per day significantly lowered risk of stroke. Evidence from RCTs and observational studies found that total fat intake was not associated with all-cause and CVD mortality, or the risk of CVD. However, high TFA intake significantly increased the risk of CVD and CVD mortality. Evidence from RCTs indicated consistent and inconsistent benefits of n-3 PUFA and n-6 PUFA for CVD prevention, respectively. Observational studies suggest the benefit of SFA for lowering stroke risk, and MUFA for lowering all-cause mortality and risk of stroke. In addition, evidence from observational studies also linked high fiber to CVD prevention and flavonoids to lowering CVD mortality. High potassium and low sodium intake also reduced the risk of stroke.

Our results, found from both RCTs and observational studies, suggest the significant benefit of healthy dietary patterns, especially the Mediterranean diet, for prevention of CVD. These findings correspond with the 2019 American College of Cardiology/American Heart Association (ACC/AHA) guidelines, which recommends healthy dietary patterns for lowering CVD [64]. Common characteristics of healthy diet patterns include a lower intake of red and processed meats, refined carbohydrates, sugar-sweetened beverages, and whole-fat dairy products; a higher consumption of fruits, vegetables, whole grains, nuts, and legumes; and a moderate consumption of alcohol. These dietary patterns are not nutrient-oriented, but rather consider a combination of multiple food groups. This approach provides several advantages. Firstly, people eat foods, not nutrients. Therefore, providing recommendation by dietary patterns is more practical in both public health and routine clinical practices than by nutrient alone [65]. Secondly, single-nutrient recommendations usually fail to consider substitutional effects and food replacement. For instance, our study found no benefit of a low-fat diet for CVD prevention, which may have resulted from the excess intake of other high-risk foods, such as refined carbohydrates and sugar to substitute the energy from fat [66,67]. Since the effect of foods on health depends on both synergistic and antagonistic interactions between multiple nutrients [68], nutrient orientation in generating dietary recommendations is not appropriate for CVD prevention.

The Mediterranean diet, among all healthy dietary patterns, has been the primary focus of previous studies. Several SRMAs of RCTs consistently found that the Mediterranean diet significantly decreased CVD mortality, CHD, and stroke risk. The key features of Mediterranean diet are a low intake of meat, with very low consumption of red and processed meat; a high intake of vegetables, fruits, nuts, legumes, cereals; and moderate intake of alcohol [69], all of which food groups had differing effects in CVD prevention. For instance, our review found that high intake of nuts, whole grains, and legumes, with a moderate intake of fish (2–4 servings/week) significantly decreased CVD mortality and CVD risk. Fish contains long-chain n-3 PUFAs that have a beneficial effect on CVD outcomes. Findings from SRs of RCTs support the hypothesis that prescribing supplement of n-3 PUFAs significantly reduces CVD risk. In addition, a recent RCT found that icosapent ethyl significantly decreases CVD risk beyond cholesterol-lowering therapy, and now has been approved by the U.S. Food and Drug Administration (FDA) for CVD prevention in high-risk patients [70].

Olive oil and extra-virgin olive oil are major sources of fat in the Mediterranean diet. Bioactive polyphenols—agents postulated to prevent CVD [71]—are only found in extra-virgin olive oil but not in common olive oil, which could explain the lack of the link between olive oil and CHD prevention in our review. In contrast, previous RCTs have suggested the significant benefit of extra-virgin olive oil in Mediterranean diet [72]. Hence, recommendation of olive oil for CVD prevention should focus on extra-virgin olive oil rather than common olive oil. Apart from bioactive polyphenols, the cardioprotective property of olive oil may come from its high content of MUFA [73]. However, results from our review show inconsistent evidence of the benefits of MUFA for CVD prevention.

Evidence of dietary patterns have usually focused on western diets, such as the Mediterranean and DASH diets. However, diets in other regions are different from those in Western countries, and it is not practical to recommend Western dietary patterns to other regions, e.g., Asian countries. Recently, the Japan Collaborative Cohort study that evaluated the effect of high Japanese food scores (i.e., high of consumption of rice, miso soup, seaweeds, pickles, green and yellow vegetables, fish, and green tea, and low consumption of beef, pork, and coffee) found that adherence to a Japanese food score may have benefits for CVD prevention [74]. However, more evidence is needed to confirm the benefits of Japanese food and other Asian dietary patterns on CVD prevention. Recommendations of dietary patterns should take into account the food culture of each region. 

SRMAs of cohort studies found no benefit of high vegetable and fruit intake, which is recommended in Mediterranean and other healthy diet patterns, on CVD outcomes. However, SRMAs of observational studies found significant benefits in high dietary fiber intake for lowering CVD mortality and CVD risk. These conflicting results might have resulted from dietary fiber consisting of vegetables, fruits, nuts, whole grains, and legumes. However, nuts, whole grains, and legumes consist of not only fiber, but also of plant protein, unsaturated fats, minerals, and phytochemicals that might be attributable for their CVD-prevention properties [75]. Evidence of CVD dietary risk factors have mainly focused on fat, which is divided into total fat intake and subtypes of fatty acids, such as saturated fat, MUFA, and n-3 and n-6 PUFA. Inconsistent results according to different outcomes were found for each subtype of fat. For instance, high saturated fat intake did not significantly increase risk of CHD and CVD mortality, but was significantly associated with lower stroke risk. Moreover, n-3 PUFA substantially decreased CVD risk, but did not decrease all-cause and CVD mortality risk. Evidence of n-6 PUFAs were conflicting for CVD mortality and CHD. Guidelines from ACC/AHD year 2019 recommends replacing saturated fat with dietary MUFA and PUFA to reduce CVD risk [64]. However, this guideline does not state clearly the type of PUFA that is beneficial for CVD prevention. 

Studies regarding some food groups and nutrients, such as vegetables, fruits, a vegetarian diet, olive oil, flavonoids, green tea, and potassium also had inconsistent findings, according to CVD outcomes. These might have resulted from the complex relationship between dietary and other lifestyle factors (e.g., physical activity), which might have confounded the findings of the previous studies.

### Strengths and Limitations

This is the first umbrella review that comprehensively reviews the evidence of diets and CVD outcomes. We considered the effect of all dietary patterns, food groups, and nutrients, according to all types of CVD outcomes, including all-cause mortality, CVD mortality, CVD, CHD, and stroke. Only systematic reviews and meta-analyses, which are in the top hierarchy of evidence, were included in the review. The quality of systematic reviews and meta-analyses were assessed using AMSTAR 2. However, our study has some limitations. Firstly, the quality of most included SRMAs were critically low. Moreover, evidence of food groups and some nutrients were from SRMAs of observational studies having more bias and confounding effects than SRMAs of RCTs. Therefore, the results from those SRMAs should be interpreted with caution. In addition, the results from the included SRMAs were qualitatively analyzed, and dietary effects on CVD outcomes cannot be exactly quantified.

## 5. Conclusions

Healthy dietary patterns, such as the Mediterranean diet, have significant beneficial effects on CVD risk. A high intake of food groups and nutrients like nuts, whole grains, legumes, and dietary fiber, and moderate intake of fish also acts to prevent CVD, while a high intake of trans fatty acids significantly increased CVD risk. With the substitutional and synergistic interactions between different food groups and nutrients, dietary recommendations for CVD prevention should be focused more on healthy dietary patterns than single food groups or nutrients.

## Figures and Tables

**Figure 1 nutrients-12-01088-f001:**
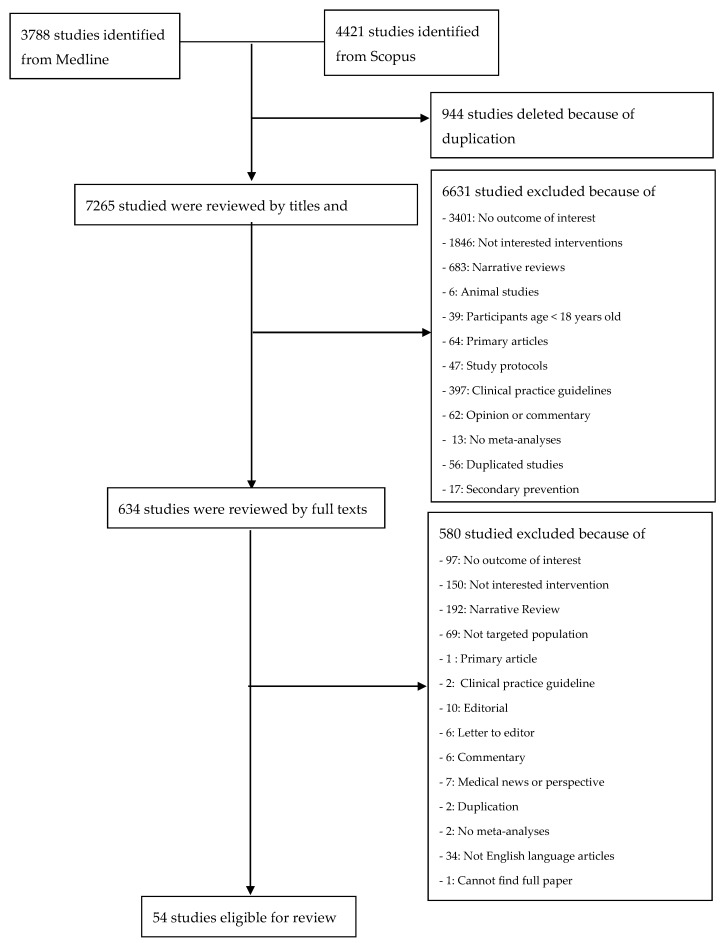
Flow chart of study selection.

**Figure 2 nutrients-12-01088-f002:**
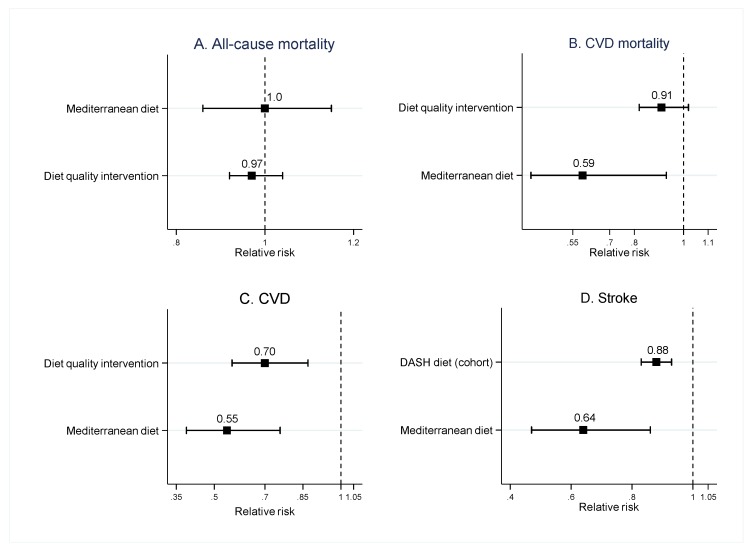
Pooled risk ratios of dietary patterns and the risk of all-cause (**A**) and cardiovascular mortality (**B**), cardiovascular disease (**C**), and stroke (**D**). Results are from systematic reviews and meta-analyses of randomized controlled trials, except for that of the Dietary Approaches to Stop Hypertension (DASH) diet, which is from observational studies. (CVD, cardiovascular diseases)

**Figure 3 nutrients-12-01088-f003:**
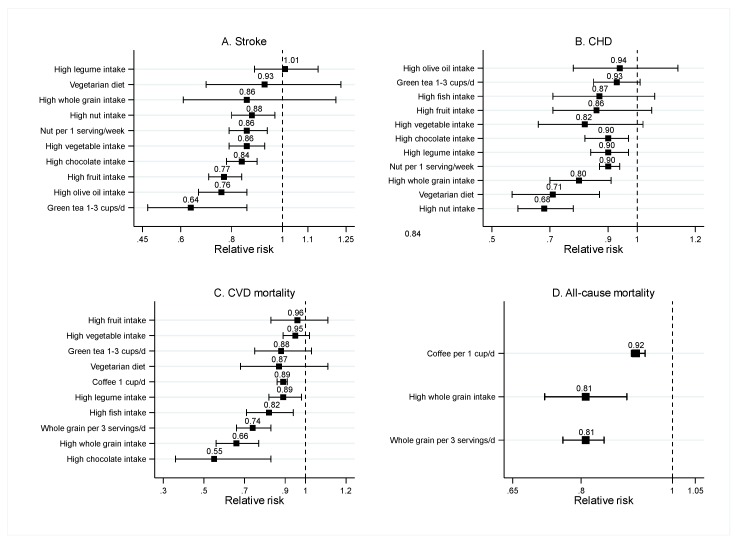
Pooled risk ratios of food groups and risk of stroke (**A**), coronary heart disease (**B**), cardiovascular mortality (**C**), and all-cause mortality(**D**). Results are from systematic reviews and meta-analyses of observational studies. (CHD, coronary heart disease)

**Figure 4 nutrients-12-01088-f004:**
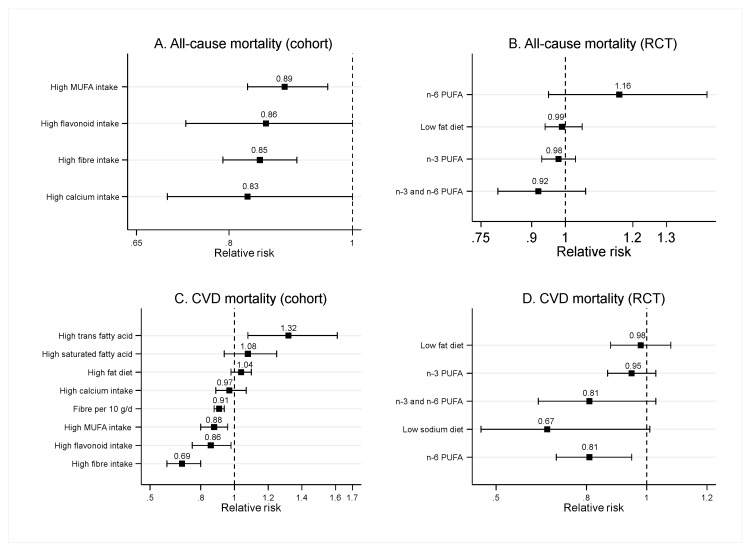
Pooled risk ratios of food nutrients and risk of all-cause and cardiovascular disease mortality. Results are from systematic reviews and meta-analyses of observational studies for (**A**) and (**C**), and systematic reviews and meta-analyses of randomized controlled trials (RCTs) for (**B**) and (**D**).

**Figure 5 nutrients-12-01088-f005:**
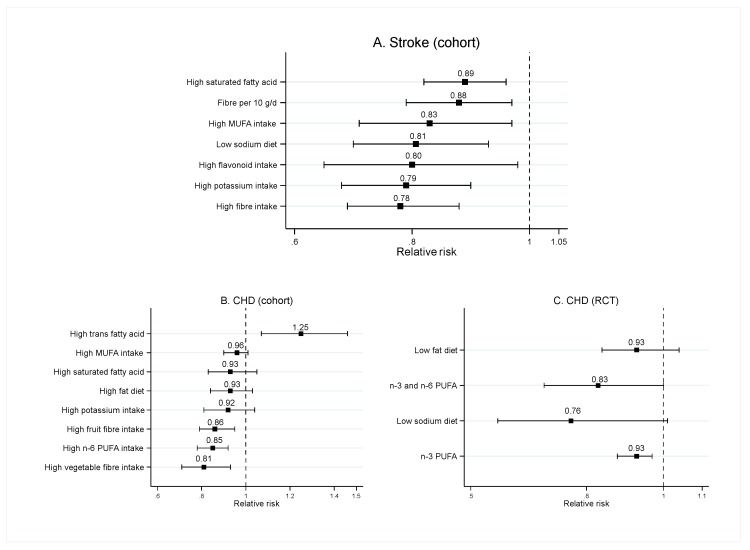
Pooled risk ratios for food nutrients and the risk of stroke and coronary heart disease. Results are from systematic reviews and meta-analyses of observational studies for (**A**) and (**B**), and systematic reviews and meta-analyses of RCTs for (**C**).

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
