# Peer review of "Dietary Factors and Risks of Cardiovascular Diseases: An Umbrella Review"

_nutrients, 2020, doi:10.3390/nu12041088_

Round 1

Reviewer 1 Report

The umbrella review by Chareonrungrueangchai et al, discusses the role of dietary factors as major risk factors for cardiovascular disease (CVD). The authors considered dietary patterns, food groups and nutrients as the major risk factors related to CVD. In this umbrella review they have performed a thorough survey of the literature using Medline and Scopus databases until June 2018 and selected 46 systematic reviews with meta-analysis (SRMA) based on their criteria to analyze and interpret the data. From the SRMA of randomized-controlled trials (RCT) and observational studies, the authors found that Mediterranean and high-quality diet can significantly lower CVD risk. They also found that low salt diet and low total fat intake did not have significant benefit for CVD prevention from SRMA of RCT studies. Moreover, their findings from different cohort studies suggest that food groups with high consumption of legumes, nuts, chocolates and vegetarian diet significantly decreased risk of coronary heart disease. At the end, they suggest substitutional and synergistic interactions between different food groups and nutrients, dietary recommendations for CVD prevention should be focused more on healthy dietary patterns than individual/single food groups or nutrients.

Overall this review is well written and summarizes the role of different dietary factors as major risk factors for the CVDs. However, I have the following suggestions and comments that can be addressed for further improvement of this review.

  1. Why have the authors only considered articles till June, 2018?

  1. Authors have discussed benefits of Mediterranean diet, DASH diet, high quality diets for CVDs. How about others diet which are specific to particular regions and ethnicities such as Asian food, Indian food, Japanese foods? It will be helpful for the readers if authors discussed such diets in the discussion of CVD risk factors.

  1. On page 9 section 3.3.1 fat intake section, line 264-266, authors have mentioned CHD mortality while referring to Figure 4C. But Figure 4C represents data related to CVD mortality (Cohort), please clarify.

Reviewer 2 Report

The authors have done a great job of searching the literature and writing an umbrella review. This type of review provide a summary of existing
research syntheses related to CVD and diet and it
presents the most scientific evidence. Currently, there are many publications that have analyzed the effect of diet with CVD and an umbrella reviwe could help us in the synthesis of results. Nevertheless, reading the article is difficult and the umbrella review should clarify the results of the published Reviews and Meta-analyzes. Authors are encouraged to focus on a specific diet issue, such as eating pattern and/or food group or nutrient. It is recommended to exclude those topics that only have one type of review.

The manuscript presents some types throughout the text. In the abstract, for example on lines 13-16-20 there is no space between the word and brackets. In the introduction and discussion, the same occurs with the square brackets of the bibliographic references, for example lines 36 and 38 or 158 and 163.

In Supplementary Table S1 and S2, references 9 and 32, the surname of the first author, Dr. González is poorly indicated. The correct name is Martínez-González.

Line 18. The type of study carried out must appear in the abstract.

Line 40. Mediterranean Diet. It is written only Mediterranean

Line 61. Indicate bibliographic reference of Umbrella review

Line 70. Indicate bibliographic reference of PRISMA

Line 113. Indicate bibliographic reference of nutrient

Line 116. Indicate bibliographic reference of diet intervention

Line 130. Numbers or leters. You should choose one. Eleven, 10, 12, and four SRMAs featured dietary....

Figura 2. The results of the studies that analyze the Mediterranean Diet should be located in the same place as the figure. In this case, in figure A, it should be below.

Line 217-219. In the 30 article it´s not indicated the mentioned results. "Both high vs low and dose response analyses of nut intake demonstrated a significantly beneficial effect on risk of CHD with pooled RRs ranging from 0.68 (95% CI: 0.59-0.78) to 0.90 (95% CI: 0.87-0.94)".

The results expressed in the text are sometimes not clarified in the figures. The authors could explain to the reviewers why they have chosen to represent one of the two results obtained in the reviews reviewed. For example, lines 229-230 "and found that high fish intake significantly reduced CHD mortality with pooled RRs ranging from 0.75 (95% CI: 0.62-0.92) to 0.82 (95% CI: 0.71, 0.94), see Figure 3B". In the Figure 3B the data shown is 0.82 (95% CI: 0.71, 0.94). Could the authors explain why, please.

Round 2

Reviewer 2 Report

The authors have clearly responded to all comments made by the reviewers.